# Imaging Methods to Quantify the Chest and Trunk Deformation in Adolescent Idiopathic Scoliosis: A Literature Review

**DOI:** 10.3390/healthcare11101489

**Published:** 2023-05-19

**Authors:** Ana San Román Gaitero, Andrej Shoykhet, Iraklis Spyrou, Martijn Stoorvogel, Lars Vermeer, Tom P. C. Schlösser

**Affiliations:** 1Master’s Medical Imaging, Utrecht University, 3584 CS Utrecht, The Netherlands; 2Department of Orthopedic Surgery, University Medical Center Utrecht, G05.228, P.O. Box 85500, 3508 GA Utrecht, The Netherlands

**Keywords:** adolescent idiopathic scoliosis, chest deformity, imaging, chest parameters

## Abstract

**Background context**: Scoliosis is a three-dimensional deformity of the spine with the most prevalent type being adolescent idiopathic scoliosis (AIS). The rotational spinal deformation leads to displacement and deformation of the ribs, resulting in a deformity of the entire chest. Routine diagnostic imaging is performed in order to define its etiology, measure curve severity and progression during growth, and for treatment planning. To date, all treatment recommendations are based on spinal parameters, while the esthetic concerns and cardiopulmonary symptoms of patients are mostly related to the trunk deformation. For this reason, there is a need for diagnostic imaging of the patho-anatomical changes of the chest and trunk in AIS. **Aim**: The aim of this review is to provide an overview, as complete as possible, of imaging modalities, methods and image processing techniques for assessment of chest and trunk deformation in AIS. **Methods**: Here, we present a narrative literature review of (1) image acquisition techniques used in clinical practice, (2) a description of various relevant methods to measure the deformity of the thorax in patients with AIS, and (3) different image processing techniques useful for quantifying 3D chest wall deformity. **Results**: Various ionizing and non-ionizing imaging modalities are available, but radiography is most widely used for AIS follow-up. A disadvantage is that these images are only acquired in 2D and are not effective for acquiring detailed information on complex 3D chest deformities. While CT is the gold standard 3D imaging technique for assessment of in vivo morphology of osseous structures, it is rarely obtained for surgical planning because of concerns about radiation exposure and increased risk of cancer during later life. Therefore, different modalities with less or without radiation, such as biplanar radiography and MRI are usually preferred. Recently, there have been advances in the field of image processing for measurements of the chest: Anatomical segmentations have become fully automatic and deep learning has been shown to be able to automatically perform measurements and even outperform experts in terms of accuracy. **Conclusions**: Recent advancements in imaging modalities and image processing techniques make complex 3D evaluation of chest deformation possible. Before introduction into daily clinical practice, however, there is a need for studies correlating image-based chest deformation parameters to patient-reported outcomes, and for technological advancements to make the workflow cost-effective.

## 1. Introduction

The most prevalent type of scoliosis is adolescent idiopathic scoliosis (AIS), a spinal curvature affecting previously healthy adolescents during the pubertal growth spurt. Scoliosis is defined as a spinal deformity with a radiographic Cobb angle of more than 10 degrees in the coronal plane. It is, however, more accurately characterized as a 3-dimensional (3D) deformity of the spine in the coronal, sagittal, and axial planes [1,2,3]. Due to the intrinsic rotation of the curvatures, significant deformation of the chest also occurs [4,5]. For primary thoracic curves, in the coronal plane, lateral bending is related to narrowing of the hemithorax on the convex side of the curve. In the axial plane, the so-called rib-hump is the most obvious component of the rib cage deformity. This is the result of axial vertebral rotation, and some intrinsic rib deformation. In the sagittal plane, the often hypokyphotic thoracic spine penetrates the thorax, leading to an ‘endothoracic hump’, narrowing of the convex airways, and alterations in the cardiac pressures.

The chest deformation in AIS can cause significant impairment in self-image, and functional impairments of the shoulders, ribs, and scapula, as well as the cardiopulmonary system, depending on the severity of the disease [6]. These patients can face an inability to exercise due to decreased chest volume and/or compliance and the impaired function of the diaphragm, which highly affects the patient’s health and self-esteem [7]. Operative procedures mainly focus on correcting the curvature of the spine, which does not necessarily result in a complete normalization of the chest deformity.

Routine diagnostic imaging is used for measuring the severity and progression of the spinal curvatures in scoliosis. The diagnostic methods currently used in the evaluation of scoliosis include conventional radiography, biplanar radiography, magnetic resonance imaging (MRI), computed tomography scans (CT), 3D ultrasound, and surface topography. In clinical practice, the diagnosis and management of scoliosis are often only based on two-dimensional images (2D), a lateral and posteroanterior radiograph [8]. Moreover, often only the deformity of the spine is considered without systematic evaluation of the patho-anatomical changes of the chest or trunk. As a result, only the coronal and sagittal projections of the spinal deformity are analyzed, excluding the axial projections, which makes it impossible to accurately assess the complex 3D deformity of the spine and other important deformities such as the chest. 

For decades, all treatment recommendations for AIS patients have been based on spinal parameters, while the esthetic concerns, cardiopulmonary symptoms and maybe even the paraspinal pain of patients are mostly related to the trunk deformation. Because of significant technological advancements, systematic evaluation of chest deformation is now becoming available for clinicians treating AIS patients. With these advances in various fields, there is a need for a comprehensive update on (innovative) methods available to systematically evaluate chest deformation in AIS.

In this narrative review, we examine the literature to explore various ways of quantifying the deformity of the chest in adolescent idiopathic scoliosis with different imaging techniques. Section 2 provides an overview of the several image acquisition techniques used. In Section 3, various relevant methods to measure the deformity of the thorax for patients with AIS are discussed. In Section 4, different image processing techniques useful for quantifying 3D chest deformity are discussed. Finally, a discussion and conclusions related to the modalities, evaluation and quantification methods are presented.

## 2. Imaging Modalities

The purpose of this section is to give an overview of the different acquisition techniques that are used for imaging the chest deformity in scoliosis, along with their advantages and disadvantages. The most widely used imaging modality is radiography, and other modalities are biplanar stereo-radiography, CT, MRI, surface topography and 3D spinal ultrasound [9]. A general overview of the modalities and their clinical parameters is presented in Table 1 and Figure 1.

### 2.1. Radiography

AIS can be diagnosed and managed effectively with radiographs for both initial diagnostic imaging evaluations and follow-up [10]. With radiography, two-dimensional images are acquired in two projections: posterior–anterior and lateral. This allows clinicians to calculate all the primarily diagnostic parameters for scoliosis, such as the curve type, the coronal Cobb angle, the apical vertebral rotation or the coronal balance [9]. Radiographs remain the mainstay in diagnosis and evaluation of scoliosis progression. Furthermore, they can show the severity of the rib hump; however, the planar projection of radiographs constrains their use in the evaluation of vertebral rotations or complex 3D deformities of the spine or chest. Repeated radiography has negative long-term consequences due to the harmful radiation the method entails [11]. Several studies have shown that AIS patients are at increased risk of breast cancer due to repeated radiographic examinations [9].

### 2.2. Biplanar Stereo-Radiography

Biplanar stereo-radiography (EOS Imaging, Paris, France) is a low-dose radiographic method that uses an ultra-sensitive multi-wire proportional chamber detector [9]. This system takes simultaneous anteroposterior and lateral 2D images and can be utilized to perform 3D reconstruction based on statistical models [12]. Similar to conventional radiography, biplanar radiography is useful for scoliosis quantification since it takes images while the patient is in an upright standing position. The images can be used to determine relationships among adjacent segments (cervical spine, pelvis, and lower limbs). Compared with conventional radiography, a biplanar radiography system is costly; however, the costs for the patient/parents are comparable, and patient throughput is generally higher, which partly counterbalances the difference in break-even costs [13]. This method reduces patient dose by 8–10 times compared with conventional radiography and by 800–1000 times compared with high-resolution 3D CT reconstructions [3]. Unfortunately, due to the high system costs involved, its availability in low- and middle- income countries is fairly limited [14]. Therefore, strategies to improve the accessibility to biplanar radiography technology are being investigated, as its significant difference in radiation dose compared to CT and radiography is a potentially strong advantage in the diagnosis of idiopathic scoliosis in pediatric patients [3].

### 2.3. CT

CT addresses the limitations of conventional radiography by providing accurate 3D reconstructions, which is crucial for visualizing the complex osseous abnormalities of scoliosis [10]. It provides excellent visualization of the spine, chest and organs within the chest (Figure 2). The main role of CT in AIS is not to monitor progression but it is sometimes used to plan surgical correction and spinal instrumentation [9]. Three-dimensional imaging brings many advantages, such as spatial imaging, detection of canal deformities and congenital spinal malformations, visualization and localization of spinal implants and the assessment of the quality of bones [11]. Although CT scans are the most frequently used clinical standards to provide an accurate 3D reconstruction for spinal measurement during surgical planning in adults, their high radiation exposure limits their use on pediatric and adolescent patients. Especially for repeated exposures during pre- and postoperative evaluations, the high dose is an important consideration. Moreover, the non-weight bearing position on the spine due to the prone scanning position can cause a considerable alteration to the scoliotic curvature. Additionally, CT scans require more time and are more expensive than spinal radiographs, limiting their use in clinical settings for AIS [15].

### 2.4. MRI

MRI is used when certain features such as pain, neurological symptoms or an atypical curvature pattern and abnormalities of the spinal canal are present in the patient’s symptoms, and for preoperative screening for neural axis abnormalities [9]. As a result, MRI is mostly interesting to use when patients suffer from functional impairment of the lungs or experience neural pain so that the canals of the nerves, for instance, can be evaluated [16]. Although MRI is mainly used to assess soft tissue rather than bone structures, its capability to obtain a 3D view of the patient allows the evaluation of metrics such as the volume of organs, which could have been compressed or deformed by the change in rib-cage morphology. It does not require ionizing radiation and it is thought to be superior in regards to the detail of the image. MRI has limitations compared with CT since CT scans are better at assessing bones and they are significantly cheaper and faster to acquire [11]. In the future, MRI-based synthetic CT may become more widely available for assessment of spinal morphology in AIS patients [17].

### 2.5. Surface Topography

Body surface topography is a photogrammetric technique that deals with the reconstruction of shapes, sizes, and mutual positions of objects based on photograms [11]. It is mostly used in research settings to assess the posterior trunk appearance. Many advantages such as non-invasiveness, safety, and quick and accurate assessment of body posture in three planes make it an attractive technique for a specific purpose in research settings and in evaluation of nonoperative treatment [9]. Although surface topography is a precise technique for assessing posture, all the tissues that are outside of the chest such as muscles, fat or skin impede the accurate visualization and detection of deformity of the structures of the chest, creating limitations because of variable accuracy.

### 2.6. 3D Spinal Ultrasound

This imaging method is rarely used and, to date, is primarily a research technique for assessment of spinal curvatures. The 3D ultrasound method was developed in an attempt to reduce radiation exposure for AIS patients since it can be used to safely monitor curve progression over time without the need for repeated radiographic examinations at short intervals [3]. Ultrasound has been shown to be able to accurately measure the coronal Cobb angle, but it is still at an early stage of development [9]. Therefore, its benefits include its low cost and absence of radiation, unlike the other imaging modalities discussed previously. Moreover, 3D ultrasound has been recently introduced for 3D bone modelling. Several studies have shown that 3D reconstructions of the posterior elements of the spine from ultrasound images are very similar to CT scan images and are not significantly affected by the posterior fat pad [18]. The method, though promising, is not without limitations. These include the inability to detect congenital anomalies of the anterior spinal column, and since only part of the posterior chest wall is included, the deformation of the chest wall is not measured by this technique.

## 3. Methods to Quantify Thoracic Deformity

Various different methods exist to measure the deformity of the thorax in patients with AIS. These methods and parameters are, however, still very advanced and primarily used in research for pulmonary function, and lung volumes prior to scoliosis surgery. Furthermore, advanced imaging including the complete chest is rarely repeated after surgery because of concerns about radiation exposure. For this reason, there is very limited evidence on the relevance of the various chest deformity parameters, especially for those that are not part of the major pediatric spine study groups. The Scoliosis Research Society glossary does not include thorax deformity measures and their respective correlations with spine deformity. Harris et al. [19] and Easwar et al. [20] provide an overview of all the parameters available, based on the anatomical aspect of the deformities. An overview of the most commonly used parameters can be found in Figure 3.

Different imaging modalities focus on different parts of the anatomy. There are four main image types from which the parameters are measured: coronal 2D images, axial 2D images, sagittal 2D images, and 3D images. Most of the parameters describe the sternum or the ribs in reference to the spine. Often the ratio of a certain distance or angle from the vertebra/sternum and the ribs to both sides is calculated. This review focuses more on the measurement technique than on anatomical details. Therefore, it was decided to divide the parameters that can be measured by structures that are used and metrics that were measured.

### 3.1. Parameters on the Relation between the Sternum and the Spine

There are several ways to measure the relative positions of the sternum to the spine. It is known that in a healthy chest, the spine and the sternum are in a central position, thus in the midsagittal anatomical plane. In persons who suffer from AIS, both the sternum and spine rotate away from the midline.
Distance measurements:Sagittal diameter and sternovertebral distance: These are two parameters for measuring the sagittal depth. The sternovertebral distance is the anterior–posterior distance between the posterior midpoint of the sternum and the anterior point of the apical vertebra [21]. The sagittal diameter is the anterior–posterior distance between the posterior midpoint of the sternum and the anterior point of the foramen. Both are measured on an axial CT scan. Takahashi et al. [22] report a significant negative correlation between the sagittal diameter and spine curvature. Measurements of the sagittal depth have been used to evaluate pre- to postoperative differences [21]. It has been shown that the sagittal diameter, as well as the sternovertebral distance, significantly decreases after spinal fusion surgery [21]. The sagittal diameter has also been used to evaluate the effect of thoracic deformity on pulmonary function [22].Vertebral translation: This is a measure of the lateral displacement between the spine and the sternum. For a healthy person, where the sternum is in front of the spine, this distance is zero. Vertebral translation measurements have been used for pre- and postoperative comparison of the thorax as well as in research for comparison with other parameters. A correlation of the vertebral translation with the coronal Cobb angle has been shown by Easwar et al. [20]. The vertebral translation was used to measure the effect of spine fusion surgery on the thoracic shape. It has also been shown that the parameter decreases significantly after spinal fusion surgery [21].Spinal intrusion ratio: This parameter represents the sagittal intrusion of the thoracic spine into the chest and has been demonstrated to be related to the size of the airways [5,23]. SIr is the ratio between the distance from the anterior vertebral column contour to the interior contour of the convex ribs and the distance from the internal sternal surface to the anterior contour of the vertebral column.Angular measurements:Angle from sternum to apical vertebra: This angle is provides information about the chest wall deformity. It is measured by drawing a lateral line through the sternum and a posterior–anterior line through the apical vertebra. In a healthy person, the line through the vertebra crosses the sternum at its center, and the angle is 90 degrees. The smaller the angle the more severe the deformity. The angle from the sternum to apical vertebra has been used to compare the pre- and postoperative shape of the thorax. Significant correlations between this angle and the coronal curve severity have been found [24].Midline deviation: This is the angle between the anterior–posterior line and a line that connects the anterior midpoint of the sternum with the posterior point of the foramen. In older studies, a significant correlation between the coronal curve severity and the midline deviation has been shown [25]. More recent studies also show that the midline deviation improves after spinal fusion surgery [21].Thoracic rotation: This is another way to measure the angle between the vertebra and the ribs [26]. It is measured between a line that connects the sternum with the anterior point of the vertebra and an anterior–posterior line through the vertebra. It has not been investigated if this parameter is correlated with the coronal curve severity [19]. It has, however, been shown that the thoracic rotation decreases significantly after spinal fusion surgery [21].

### 3.2. Parameters on the Relation between the Ribs and the Spine


Distance measurements:Posterior hemithoracic symmetry ratio: This ratio can be measured on an axial image of the thorax. A line is drawn through the anterior tips of the ribs where they are connected to the vertebra. Then the distance is measured from the connection point to the vertebra and the border of the hemithorax is measured. In a symmetrical thorax, the distance is equal in both directions. The ratio of the distances is 1. The more the thorax is deformed, the bigger the difference in the distances and the more the ratio is different from 1 [26].Apical vertebral body rib ratio and sternum rib ratio: An anterior–posterior radiograph is sufficient to measure the apical vertebral body rib ratio. From the apical body, the distance to the lateral borders of the ribs is measured in both directions [27]. While Kulko et al. describe it for an anterior–posterior radiograph, Mao et al. measure it on an axial reconstruction. Both parameters have been used to evaluate the pre- to postoperative shape of the thorax.Angular measurements:Rib hump and chest wall angle: The rib hump and the chest wall angle are both an indicator of rib rotation. The chest wall angle is measured on the anterior part of the rib [24], whereas the rib hump is measured on the posterior parts of the ribs or sometimes also on Moiré topographs of the back [22]. Aaro et al. [25], Kuklo et al. [27] and Takahashi et al. [22] define the rib hump or the rib hump index slightly differently. However, the measured structures remain the same. On the posterior parts of the ribs, the distance in the anterior–posterior direction between the two prominences of the ribs is measured. This can be performed on a lateral radiograph [27] or on an axial CT scan [22]. The chest wall angle is a similar measurement but for the anterior rib prominences instead of the posterior rib prominence. In addition, instead of measuring a distance, the angle of the line connecting the prominences and the lateral plane is measured. Both the rib hump and the chest wall angle have significant correlations with the spinal curvature [22,24,25]. In addition, these parameters have been used for intraoperative comparisons of the thorax. It has been shown that the effect of surgery on the chest wall shape is very hard to predict [24].Inclination angle [28]: This angle is measured by connecting the tips of one rib at the sternum and at the vertebra with a line. The angle from this line to the horizontal plane is the inclination angle. Schlager et al. [28] concluded that the rib hump in AIS could result from a more ventral position of the ribs on the concave side rather than on the bulging position of the ribs on the convex side.


## 4. Image Processing Techniques for Chest Deformity Analyses

In order to evaluate scoliosis and deformities, various image processing techniques are used, such as segmentation, 3D reconstruction techniques and deep learning.

### 4.1. Segmentation

Segmentation of the various structures in the chest can be performed manually, semi-automatically or fully automatically. For the spine, Thalenga et al. [29] described a segmentation technique on radiographs to determine the curvature of the spine and several algorithms were used to aid in spinal extraction and find anatomical landmarks for evaluation of scoliosis severity.

To assess chest deformation, Shantanu et al. presented an automated segmentation technique of the ribs based on CT images. In this technique, the first step includes the deletion of peripheral artefacts, such as external air and the skin. Next, the image becomes binarized, by applying a threshold filter at each CT voxel. After applying some morphological operations and taking into consideration that in the upper thoracic region, the rib density (number of ribs in a specific area) is higher and that the ribs are smaller than in the middle thoracic region, the compactness factor (which gives information about the density of the ribs) of the inner contour of the peripheral fat region on each slice is used to differentiate between the upper thoracic region and the rest of the lower thorax, as well as the abdomen. To finalize this method, an estimation procedure of the rib structure is applied. The proposed method is limited by the use of empirically selected parameters from a CT images subset [30].

### 4.2. D Reconstruction Techniques

Many imaging modalities acquire a 2D representation of the body. However, from these images, it is almost impossible to evaluate the 3D deformation of the chest. That is why previous research studies have focused on visualizing the spine, ribs and thoracic area in 3D from modalities such as biplanar radiographs or CT.

For biplanar radiographic images, several publications propose methods to reconstruct a 3D image of the chest. Similar to the spine reconstructions, this needs user input to determine anatomical landmarks. This process is prone to differences in image interpretation and variability between users, and is time-consuming. Vergari et al. [31] developed a fast method for the identification of the ribs and provided a reliable reconstruction solution to calculate chest parameters in mild and severe scoliosis, compared with the manual selection of specific landmarks on all ribs (Figure 4). The method utilizes custom software that zooms in on different control rib positions in the frontal view. The user can select different sets of pixels on the ribs to identify the rib segments. These sets of pixels are used to automatically estimate the position of the rib’s vertebral joint, distal tip and the most lateral point. In the lateral view, dots are placed on the most posterior corners of visible ribs. Subsequently, the software computes a spline through the dots, and the combination of landmarks on the lateral and anterior–posterior images enables the software to reconstruct the chest in 3D. The semi-automated software reduces the time of image reconstruction to less than 2 min. However, training is required in order to select structures in the images and operate the software [31]. The 3D reconstructions from biplanar stereo-radiography have proven to be a reliable method to measure chest parameters, and Machino et al. [32] have demonstrated that 3D reconstructions correlate with pulmonary functions. Jovilet et al. [33] compared several parameters to CT, finding a difference of 0.1% in measurements of the area of the chest and 6.3% in the maximum width of the chest. Nevertheless, to date, 3D reconstruction of the chest using biplanar stereo-radiography is not fully established.

While various imaging software packages are available for 3D chest reconstructions of CT images, an interesting method, not yet used for scoliosis, is the rib centerline extraction method. Lenga et al. [34] proposed a method that uses a full convolution neural network combined with a centerline extraction algorithm to evaluate the ribs in CT images. First, the convolutional neural network (CNN) generates probability maps for detecting the first, last and intermediate ribs. Next, a centerline detection algorithm is applied to the probability maps through the depth of the image to iteratively trace the rib lines. The results showed an error in the rib centerlines of only 0.787 mm. Due to the intrinsic factors of these ribs, this method shows a higher error in the distance for the first and twelfth rib compared with the others. This method can be used in computing the orientation of the ribs. The center line can be used to extract coordinates and perform calculations on rib deformity.

### 4.3. Deep Learning Techniques

Deep learning techniques can be used for image recognition, classification, detection and prediction in AIS and are rapidly reducing the time-consuming work required of experts. In general, research into deep learning techniques to measure chest parameters remains scarce, but some research has focused on using deep learning to aid in quantifying chest deformity as well as measurement of the spine [35,36]. Deep learning reconstruction is not yet widely used. The authors expect, however, that with the development of radiation-free, MRI-based synthetic CT, further research into automatic segmentation of 3D images will occur. Such research will rely on neural networks and include patient outcomes, and importantly, with the availability of more computational power, deep learning techniques will be applied in the clinical practice for AIS patients.

## 5. Conclusions

Because the main consequences of AIS for patients are not related to the spine, but to the deformation of the trunk and mostly the chest, the methods for quantification of chest deformity need to be further improved. This narrative review provides an overview of imaging modalities, methods and image processing techniques for assessment of chest deformation in AIS. Due to its narrative design, this overview prone to selection bias; however, this design permits a wider and more inclusive overview of technological advancements in imaging methods to quantify chest deformation in AIS. After evaluating different modalities, metrics and processing techniques, several aspects to focus on in further research into the quantification of the 3D deformation of the chest can be identified. The risk of bias in the selection of individual studies remains, however, a limitation of this study. In general, it is very hard to decouple spinal and thorax deformities. Especially spinal rotation affects many parameters. It is sometimes difficult to determine if it is actually rib deformity, rib displacement or spinal deformity that is measured [19]. AIS patients often come into contact with radiography numerous times and suffer from being exposed to high doses. Imaging modalities which do not expose patients to ionizing radiation exist; however, as yet, they less suitable for rib-cage visualization. Biplanar stereo-radiography is a technique which allows a reduced dose compared with radiography and CT. The process of 3D reconstruction of the chest, however, is still time consuming. However, it is evident that at the moment, CT is more readily available and can provide detailed 3D reconstructions of the chest as well as the organs within the chest. The metrics currently used in quantifying deformation do not yet give a clear overview of how the chest is deformed in 3D, and do not give a clear indication about whether the ribs are deformed. In image analysis, deep learning is entering the field of chest parameter measurement and has already proven to be able to outperform experts in CT images. Deep learning reconstruction is not yet widely used. However, due to the promising results it can be assumed that deep learning techniques will become more common. Image processing techniques such as segmentation are available and are used to extract the spine from radiographic images. However, this is more error-prone compared with automated 3D reconstructions from biplanar stereo-radiography or CT. Although 3D reconstruction from biplanar stereo-radiography is not fully established yet in clinical settings, it is able to compete with CT on measurements of the chest parameters. Nevertheless, it is typically easier to use fully automated techniques to evaluate chest parameters from CT than techniques needing more input from trained experts.

In conclusion, because of significant technological advancements, systematic evaluation of chest deformation is becoming available for clinicians treating AIS patients. There is, however, a need for studies correlating image-based chest deformation parameters to patient-reported outcomes. Furthermore, in contrast to the conventional 2D radiography-based spinal parameters, 3D evaluations are more complex and require more time for quantification of anatomical parameters. Further advancements in automatic 3D image evaluations are first needed before complex chest deformity evaluations can be introduced into daily practice.

## Figures and Tables

**Figure 1 healthcare-11-01489-f001:**
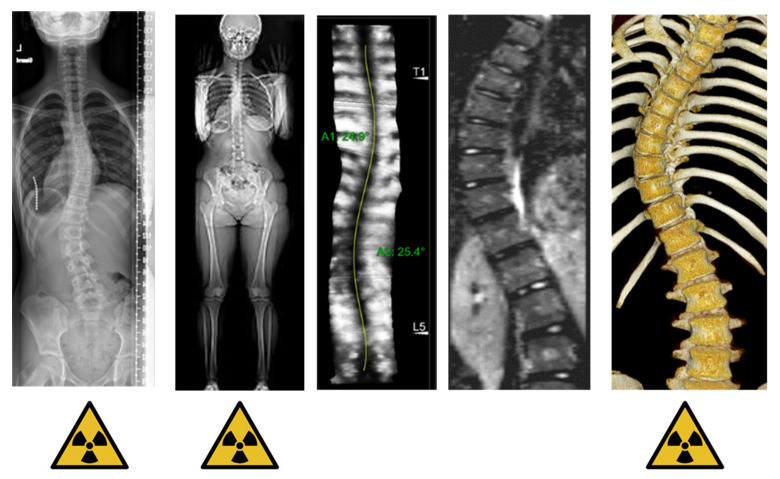
Overview of the most common imaging modalities: radiography, biplanar radiography, spinal ultrasound, MRI and CT.

**Figure 2 healthcare-11-01489-f002:**
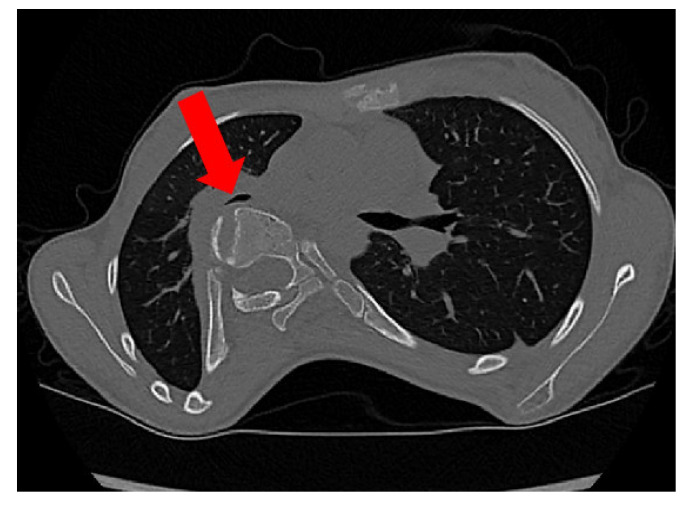
An axial CT reconstruction that demonstrates the intrusion of the spine into the right hemithorax. The red arrow highlights the narrowing of the right bronchus intermedius.

**Figure 3 healthcare-11-01489-f003:**
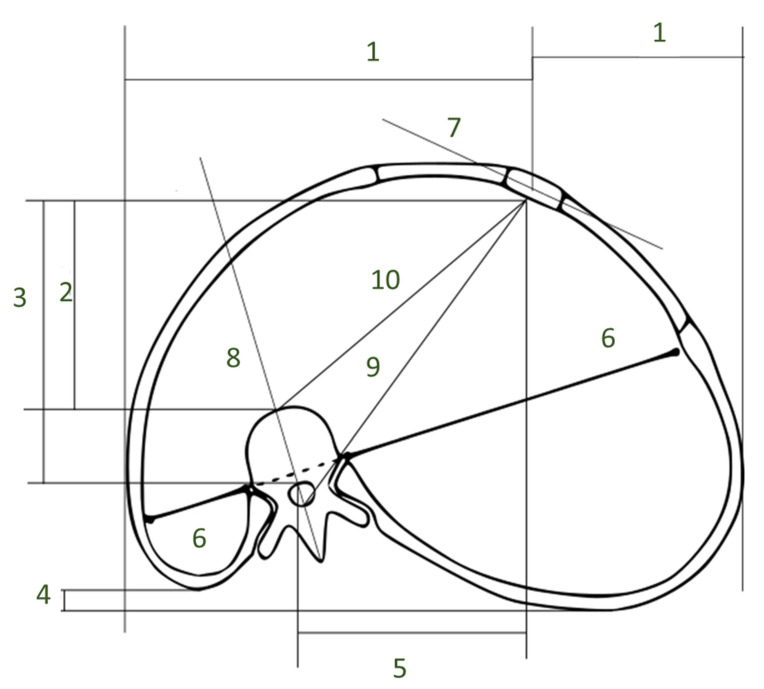
Overview of rib cage parameters that can be measured on an axial CT scan. A legend can be found in Table 2.

**Figure 4 healthcare-11-01489-f004:**
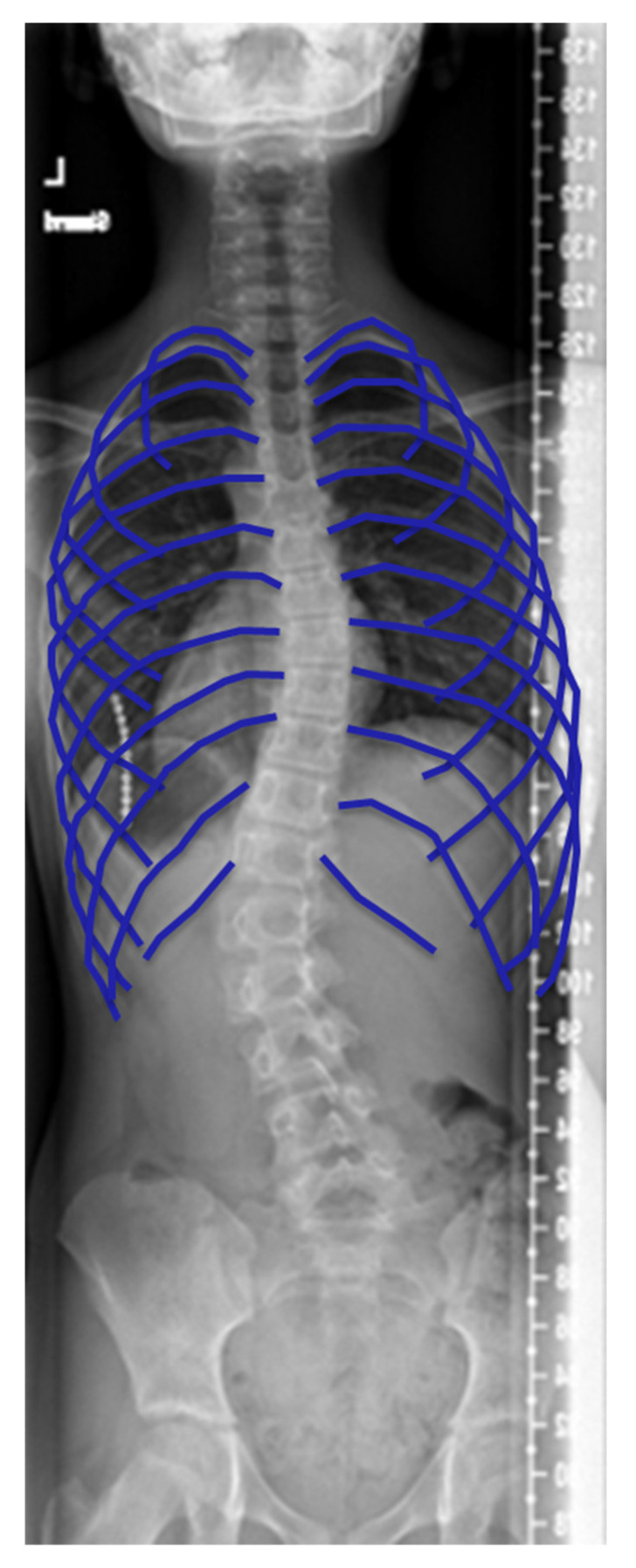
A reconstruction of the chest for radiography, similar to the technique performed by Vergari et al. using coloring of the ribs [31].

**Table 1 healthcare-11-01489-t001:** A general overview of the modalities and their clinical parameters available for imaging of the chest in AIS.

Modality	Radiation Exposure	Assessment of Rib Deformity	Costs	Principle Clinical Role
Radiography	Low	Planar evaluation of rib deformity.	Low	Evaluation of scoliosis severity and progression during follow-up.
Biplanar stereo-radiography	Ultra-Low	Deformity evaluation of the bone structures. Possibility for 3D reconstruction.	Low	Evaluation of scoliosis severity and progression during follow-up.
Surface Topography	None	External contour of the trunk. No direct insight into the rib cage deformity.	Medium	Mostly used in research settings for objective trunk deformity assessment.
3D Ultrasound	None	Visualization of spinal posterior elements. Only first dorsal part of rib deformity included.	Low	Mostly used in research settings for screening for scoliosis and evaluation of scoliosis severity.
CT	High	High accuracy of rib cage and internal organ structures. Possibility for 3D reconstructions and automatic segmentation.	High	Sometimes used for visualizing complex osseous abnormalities, for preoperative planning or as input for spinal navigation surgery.
MRI	None	Mainly used to assess soft tissue rather than bone structures.	High	Evaluation for abnormalities of the spinal cord.

**Table 2 healthcare-11-01489-t002:** Description of parameters shown in Figure 3.

Sagittal diameter	Length 3
Sternovertebral distance	Length 2
Vertebral translation	Length 5
Angle from sternum to apical vertebra	Angle between 7 and 8
Midline deviation	Angle between 9 and anterior posterior plane
Thoracic rotation	Angle between 8 and 10
Posterior hemithoracic symmetry ratio	Ratio of both 6
Sternum rib ratio	Ratio of both 1
Rib hump	Length 4

## Data Availability

Not applicable.

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
