# Peer review of "Imaging Methods to Quantify the Chest and Trunk Deformation in Adolescent Idiopathic Scoliosis: A Literature Review"

_healthcare, 2023, doi:10.3390/healthcare11101489_

Round 1

Reviewer 1 Report

The authors reviewed the current ways to quantify deformity in AIS patients and summarized the methods for measurements and finally discussed the image processing process. 

This review provides detailed information for health providers and family members of patients. However, the authors should pay more attention to their minor errors and read the manuscript more carefully. For example, they should make sure references are correctly included when using ref manager software (Lines 74 and 190). Also, in the method section, please be consistent with the numbering system, otherwise, the description of detailed measurements is very hard to follow. Please also check subtitles and make sure they are not repetitive (e.g. line 203 and 245) or change levels of indentation, so it becomes more clear. For titles with numbers (3D), please make sure they are not automatically edited. 

Deep learning looks promising but may not be widely used. Maybe the authors could also suggest ways for it to become more popular or discuss what else is needed for better usage.

Author Response

The authors reviewed the current ways to quantify deformity in AIS patients and summarized the methods for measurements and finally discussed the image processing process. This review provides detailed information for health providers and family members of patients. However, the authors should pay more attention to their minor errors and read the manuscript more carefully. For example, they should make sure references are correctly included when using ref manager software (Lines 74 and 190).

Authors’ response: Thank you very much for the contributive comments. We would like to apologize for the incorrect references. These were references to tables, inserted by the editorial team of the journal.

Changes in manuscript: correct references to relevant tables and figures.

Also, in the method section, please be consistent with the numbering system, otherwise, the description of detailed measurements is very hard to follow.

Authors’ response: We would like to apologize for the incorrect numbering system. This was the result of the conversion by the editorial team of the journal.

Changes in manuscript: numbering system has been updated

Please also check subtitles and make sure they are not repetitive (e.g. line 203 and 245) or change levels of indentation, so it becomes more clear.

Authors’ response: This repetition was by purpose. The subtitles were not edited properly in the version send to the reviewers.

Changes in manuscript: Subtitles and numbering has been updated.

For titles with numbers (3D), please make sure they are not automatically edited. 

Authors’ response: Thank you.

Changes in manuscript: changed accordingly.

Deep learning looks promising but may not be widely used. Maybe the authors could also suggest ways for it to become more popular or discuss what else is needed for better usage.

Authors’ response: We agree that deep learning techniques are not yet widely used. As suggested, we are convinced that deep learning techniques will become more popular with time, and become part of the daily clinical practice.

Changes in manuscript: We added discussion on deep learning, in the deep learning paragraph.

Reviewer 2 Report

This paper is a well-organized review article on the imaging diagnostic methods for adolescent idiopathic scoliosis.In this paper, the authors describe the advantages and disadvantages of various imaging diagnostic methods for the early detection of adolescent idiopathic scoliosis and the changes in its three-dimensional structure. The authors also provided a well-organized description of the method for measuring three-dimensional quantitative changes in AIS.

Nevertheless, I will add some comments.

In introduction, explain the rationale for the review in the context of what is already known.

In method, indicate if a review protocol exists, if and where it can be accessed (such as a web address).  In the description of some measuring and imaging processing techniques of 3-D spine deformation, there are errors in the paragraph arrangement.

In discussion, discuss limitations at this study and outcome level (such as risk of bias), and at review level such as incomplete retrieval of identified research, reporting bias.

In conclusion,  a general interpretation of the results in the context of other evidence, and implications for future research.

Thank you.

Author Response

This paper is a well-organized review article on the imaging diagnostic methods for adolescent idiopathic scoliosis.In this paper, the authors describe the advantages and disadvantages of various imaging diagnostic methods for the early detection of adolescent idiopathic scoliosis and the changes in its three-dimensional structure. The authors also provided a well-organized description of the method for measuring three-dimensional quantitative changes in AIS. Nevertheless, I will add some comments.

In introduction, explain the rationale for the review in the context of what is already known.

Authors’ response: We would like to thank the reviewer for the comments and suggestions for improvement. We agree that the rationale and context can be more clearly described in the introduction section.

Changes in manuscript: line 82-88 the rationale was added

In method, indicate if a review protocol exists, if and where it can be accessed (such as a web address).  In the description of some measuring and imaging processing techniques of 3-D spine deformation, there are errors in the paragraph arrangement.

Authors’ response: This was a narrative, non-systematic review. Unfortunately, the subtitles/paragraphs were not edited properly in the version send to the reviewers.

Changes in manuscript: The design was more explicit described in the abstract. Paragraph arrangement was updated.

In discussion, discuss limitations at this study and outcome level (such as risk of bias), and at review level such as incomplete retrieval of identified research, reporting bias.

Authors’ response: We agree that the limitations should be included in the discussion.

Changes in manuscript: risk of bias and selection bias by the narrative design (line 445-451) were added.

In conclusion,  a general interpretation of the results in the context of other evidence, and implications for future research. 

Authors’ response: we agree.

Changes in manuscript: a general interpretation and implications are added (line 474-480).

Reviewer 3 Report

The paper provides a comprehensive and up-to-date review of imaging techniques used to evaluate chest deformity in adolescent idiopathic scoliosis patients. The review is current and exhaustive, which makes it a valuable resource for professionals interested in this field. I strongly recommend its publication without any reservations, as it will undoubtedly save time and effort for those seeking relevant information.

Author Response

The paper provides a comprehensive and up-to-date review of imaging techniques used to evaluate chest deformity in adolescent idiopathic scoliosis patients. The review is current and exhaustive, which makes it a valuable resource for professionals interested in this field. I strongly recommend its publication without any reservations, as it will undoubtedly save time and effort for those seeking relevant information.

Authors’ response: Thank you very much for reviewing our manuscript.

Changes in manuscript:-

Reviewer 4 Report

1- Abstract is needed to answer the classical question (Why, How, What).

2-It  is very  efficient to  compared  present work with  other references

3-It is very important to checking the accuracy of this technique and calculate the percent of error

4-The results of parent work is needed to more discussion

5- Conclusion is needed to rewriting in order to focusing on the more important results

Author Response

1- Abstract is needed to answer the classical question (Why, How, What).

Authors’ response: Thank you very much for the comment.

Changes in manuscript: We added a structure to the abstract

2-It  is very  efficient to  compared  present work with  other references

Authors’ response: Thank you for the comment.

Changes in manuscript:

3-It is very important to checking the accuracy of this technique and calculate the percent of error

Authors’ response: The technique is not specified.

Changes in manuscript: A general discussion, however, on the relevance of the techniques was added.

4-The results of parent work is needed to more discussion

Authors’ response: There is no reference to parent work in this manuscript. We believe it does not fit well into the scope of the current imaging overview.

Changes in manuscript:-

5- Conclusion is needed to rewriting in order to focusing on the more important results

Authors’ response: Thank you.

Changes in manuscript: changed according to the reviewers suggestion.

Reviewer 5 Report

Thank you for submitting your manuscript on imaging methods for chest and trunk deformity in adolescent idiopathic scoliosis (AIS). Unfortunately I have significant concerns regarding its significance and whether it warrants publication. These concerns include:

(1). Your manuscript is complex and without much clinical relevance. Spinal measurements in AIS are principally for the magnitude of spinal deformity, determining treatment, and following the results of treatment (non-operative and operative). As such I was struck by the major question of who is the target audience for your manuscript? Certainly not orthopaedic surgeons or radiologists who are very familar with most of the more common techniques. The information will be of limited interest to pediatricians or pediatric pulmonologists who typically do not order or measure radiographs. Measurements of the chest wall and trunk are rarely performed clinically as the major concern is the deformities affect on pulmonary functions which are easier to determine by pulmonary function testing techniques.

(2). Your manuscript is not well balanced. The first part (p.1-8) is basic information that is well known to those physicians providing care for AIS. The 2nd part (p. 8-11) is research based. In my career I performed over 3000 pediatric operative spine cases and have never used any of these latter measurements. I have been aware of them but they were not clinically relevant.

(3). Overall, your manuscript needs extensive editing. The English can be improved. You used jargon terms such as X-ray, Cobb angle, and Gold Standard. There was also misrepresentation when you stated that biplaner images are of high cost. While the EOS system is expensive to purchase and maintain the cost to patients is not. Billing and reimbursement are limited to the same as PA and lateral plain radiographs. CT scans have largely been replaced by EOS and MRI.

Author Response

Thank you for submitting your manuscript on imaging methods for chest and trunk deformity in adolescent idiopathic scoliosis (AIS). Unfortunately I have significant concerns regarding its significance and whether it warrants publication. These concerns include:

(1). Your manuscript is complex and without much clinical relevance. Spinal measurements in AIS are principally for the magnitude of spinal deformity, determining treatment, and following the results of treatment (non-operative and operative). As such I was struck by the major question of who is the target audience for your manuscript? Certainly not orthopaedic surgeons or radiologists who are very familar with most of the more common techniques. The information will be of limited interest to pediatricians or pediatric pulmonologists who typically do not order or measure radiographs. Measurements of the chest wall and trunk are rarely performed clinically as the major concern is the deformities affect on pulmonary functions which are easier to determine by pulmonary function testing techniques.

Authors’ response: Thank you very much for the contributive comments. We completely agree that pediatric deformity surgeons will be familiar with most radiographic, spinal parameters included in this manuscript. We believe, however, that the target audience of this narrative review in the Healthcare Journal is a broader readership, including (surgical and non-surgical) clinicians treating patients with AIS, radiologist as well as scoliosis researchers, pediatricians, pulmonologists, image processing engineers. For this reason,we also include some basic information on AIS diagnostic imaging and parameters.

Changes in manuscript:-

(2). Your manuscript is not well balanced. The first part (p.1-8) is basic information that is well known to those physicians providing care for AIS. The 2nd part (p. 8-11) is research based. In my career I performed over 3000 pediatric operative spine cases and have never used any of these latter measurements. I have been aware of them but they were not clinically relevant.

Authors’ response: For decades, all treatment recommendations for AIS patients are based on spinal parameters, and therefore, we understand that a very experienced surgeon did never use any of the chest measurements in daily practice. The esthetic concerns, cardiopulmonary symptoms and maybe even the paraspinal pain of AIS patients are, however, mostly related to the trunk deformation. Because of significant technological advancements, systematic evaluation of chest deformation is now becoming available for clinicians treating AIS patients. We believe, however, that there is a need for a comprehensive update on (innovative) methods available to systematically evaluate the chest deformation in AIS for (surgical and non-surgical) clinicians treating patients with AIS

Changes in manuscript: The rationale for this study was added to the introduction, and discussion on clinical relevance was added to the discussion/conclusion section.

(3). Overall, your manuscript needs extensive editing. The English can be improved. You used jargon terms such as X-ray, Cobb angle, and Gold Standard. There was also misrepresentation when you stated that biplaner images are of high cost. While the EOS system is expensive to purchase and maintain the cost to patients is not. Billing and reimbursement are limited to the same as PA and lateral plain radiographs. CT scans have largely been replaced by EOS and MRI.

Authors’ response: Thank you for the comment. We agree that jargon is used. We replaced X-ray by radiography, but needed to keep the term ‘Cobb angle’ because of it’s wide use. We agree that the section on biplanar radiography misrepresents the modality and that CT is not often used in AIS. We changed it according to the suggestions of the reviewed.

Changes in manuscript: Throughout the manuscript the jargon, and costs for biplanar radiography have been replaced.

Round 2

Reviewer 5 Report

Thank you for revising your manuscript. While slightly improved I still have concerns regarding publication. My concerns regarding the revision and the overall manuscript include: 

(1). The main improvement was the presentation of the six imaging modalities. These were more clearly defined and easier to understand. Nevertheless, surface topography and 3D ultrasound are sparsely used and are primarily a research technique. They may have some clinical applications in the future. 

(2). The methods used to quantify thoracic deformity are still very advanced and used principally in research for lung volumes and pulmonary functions prior to surgery. As such, they are rarely used including major pediatric spine study groups. They are not even determined in the Pediatric Spine Study Group. They are limited to institutions with specific interests. A major problem is that they need to be repeated postoperatively to determine any efficacy. This may lead to increase exposure of ionizing radiation. Also, they would rarely be used in AIS because of the older age and maturity of these patients. Their main applicability would be in younger patients with early onset scoliosis which have the greatest potential for pulmonary compromise.

(3). I still do not understand deep learning techniques. All six imaging techniques main functions are the assessment of the spinal deformity and its response to treatment. Quantitating thoracic deformity is primarily of importance in determining space available for the lung and pulmonary function. These latter factors are of limited or any interest to most physician and surgeons involved in the care of patients with AIS. They are research tools of limited clinical applicability except perhaps in early onset scoliosis which had the greatest potential for pulmonary compromise. If ultimately accepted I would strongly suggest this portion of the manuscript be significantly reduced to the most basic information or entirely eliminated. This is much too complex for even a broad readership.

Author Response

We would like to thank reviewer #5 for reading our revised manuscript (healthcare-2341098 entitled " Imaging Methods to Quantify the Chest and Trunk Deformation in Adolescent Idiopathic Scoliosis: A Literature Review") and the contributive feedback.  

Reviewer 5:
Thank you for revising your manuscript. While slightly improved I still have concerns regarding publication. My concerns regarding the revision and the overall manuscript include:

(1). The main improvement was the presentation of the six imaging modalities. These were more clearly defined and easier to understand. Nevertheless, surface topography and 3D ultrasound are sparsely used and are primarily a research technique. They may have some clinical applications in the future.
Authors’ response: We agree completely with the reviewer that surface topography and 3D spinal ultrasound are not widely adopted techniques, and are mostly used for research purposes. Furthermore, our experience is that they are more applied in centers where mostly conservative scoliosis treatment is applied.
Changes in manuscript: these techniques were removed from the abstract (line 34). Limitations were more clearly highlighted in the manuscript.

(2). The methods used to quantify thoracic deformity are still very advanced and used principally in research for lung volumes and pulmonary functions prior to surgery. As such, they are rarely used including major pediatric spine study groups. They are not even determined in the Pediatric Spine Study Group. They are limited to institutions with specific interests. A major problem is that they need to be repeated postoperatively to determine any efficacy. This may lead to increase exposure of ionizing radiation. Also, they would rarely be used in AIS because of the older age and maturity of these patients. Their main applicability would be in younger patients with early onset scoliosis which have the greatest potential for pulmonary compromise.
Authors’ response:
- We agree that thoracic deformity parameters are still seldomly applied in daily clinical practice. While it is generally accepted that the primary aim of early onset scoliosis treatment is focused on development and growth of the chest, chest morphology is rarely evaluated. We are completely aware that the major pediatric spine study groups, including the Pediatric Spine Study Group, rarely perform quantifications of chest deformity (even for research purposes). In our opinion, however, this can be partly explained because the relevance has, and the most valuable chest deformity parameters have not been defined. Another reason is that, as of yet, 3D images including the complete chest are not part of standard follow-up of pediatric spine deformity patients. With modern, non-ioninzing imaging technique, such as BoneMRI, this is becoming available in the near future. We completely agree that this should NOT lead to increase exposure of ionizing radiation, unless there is clear evidence of value for the scoliotic patient population.
- In terms of defining the relevant parameters, unfortunately, the Scoliosis Research Society glossary does not include thorax deformity measures and their respective correlations with spine deformity. However, as included in the quantification section, a comprehensive review on relevant parameters has been performed in 2015 (including a Pediatric Spine Study Group member). In line with this review, we also believe that three-dimensional imaging methods and deformity characterization are essential in guiding surgical restoration of abnormal spine curvature as well as thoracic shape and function.
Changes in manuscript: the limitations and the fact that the parameters are rarely used by major pediatric study groups is added: “Various different methods exist to measure the deformity of the thorax for pa-tients with AIS. These methods and parameters are, however, still very advanced and primarily used in research for pulmonary function, and lung volumes prior to scoliosis surgery. Furthermore, advanced imaging including the complete chest is rarely re-peated after surgery because of concerns of radiation exposure. For this reason, there is very limited evidence on the relevance of the various chest deformity parameters and those or not part of the major pediatric spine study groups.  The Scoliosis Research Society glossary does not include thorax deformity measures and their respective cor-relations with spine deformity. Harris et al. [19] and Easwar et al. [20] provide an overview of all parameters available, based on the anatomical aspect of the deformities”

(3). I still do not understand deep learning techniques. All six imaging techniques main functions are the assessment of the spinal deformity and its response to treatment. Quantitating thoracic deformity is primarily of importance in determining space available for the lung and pulmonary function. These latter factors are of limited or any interest to most physician and surgeons involved in the care of patients with AIS. They are research tools of limited clinical applicability except perhaps in early onset scoliosis which had the greatest potential for pulmonary compromise. If ultimately accepted I would strongly suggest this portion of the manuscript be significantly reduced to the most basic information or entirely eliminated. This is much too complex for even a broad readership.
Authors’ response:
- We agree the quantification of thoracic deformity is primarily of importance in determining the morphology of the lungs and airways, and that deep learning techniques are research tools of very limited clinical applicability to date and that the paragraph can be shortened.
- We do not completely agree with the reviewer that the chest deformity analyses would be rarely be used in (operative) AIS because of the older age and maturity of these patients. Multiple studies (Newton et al. 2005, Farrell et al. JBJS 2021) have demonstrated that >1/3rd of the AIS population has a percent predicted FVC <65%. While the main applicability would definitely be in younger patients with early onset scoliosis which have the greatest potential for pulmonary compromise, we are convinced that there is also a place in the treatment of severe scoliosis in adolescents.
Changes in manuscript: the section has been significantly reduced to basic information.
